# Stress Reduction in Alpaca (*Vicugna pacos*) Mange Management

**DOI:** 10.3390/vetsci11110587

**Published:** 2024-11-20

**Authors:** Marilena Bolcato, Mariana Roccaro, Filippo Maria Dini, Arcangelo Gentile, Angelo Peli

**Affiliations:** 1Department of Veterinary Medical Sciences, Alma Mater Studiorum University of Bologna, Via Tolara di Sopra, 50, 40064 Bologna, Italy; filippomaria.dini@unibo.it (F.M.D.); arcangelo.gentile@unibo.it (A.G.); 2Department for Life Quality Studies, Alma Mater Studiorum University of Bologna, Corso D’Augusto, 237, 47921 Rimini, Italy; mariana.roccaro2@unibo.it (M.R.); angelo.peli@unibo.it (A.P.)

**Keywords:** alpaca, mixed mange, stress, management, treatment, animal welfare

## Abstract

Stress and skin issues, like itching, are closely connected. Alpacas are sensitive animals and frequently suffer from mange, which is an especially stressful disease. Moreover, veterinary procedures necessary for its treatment can unintentionally increase stress and worsen the disease. This case report emphasizes the role of stress in the development, management, and treatment of skin conditions. Three alpacas introduced to an animal theme park developed skin problems after three months. They showed mild itching, hair loss, and thickened skin lesions on their ears, belly, and limbs. Skin tests showed the presence of *Sarcoptes scabiei* and *Chorioptes bovis* mites. The alpacas were treated with subcutaneous injections four times, once a week. Weekly check-ups monitored their health condition and stress, which included increased alertness, running, jumping, vocalizations, accelerated heart rate, and breathing. Despite the test results being still positive, due to the ongoing stress, treatment was suspended after 21 days, and the disease was monitored remotely. A month later, the tests were negative, and after three months, all alpacas had fully recovered with hair regrowth. This showed that managing stress is crucial for effective disease treatment and overall well-being in alpacas, making it a key aspect of veterinary care.

## 1. Introduction

Veterinary practices are widely recognized as sources of stress in companion animals [1,2,3,4]. Several studies indicate that capture, restraint, and handling alone are a source of considerable stress in livestock [5,6,7]. This is even more true for non-domesticated prey species, particularly ungulates, where fear, restraint, and pain associated with veterinary intervention and treatment may aggravate the illness [8,9,10]. The popularity of alpacas (*Vicugna pacos*) as pets has been increasing in Northern Europe [11,12,13,14]. Despite their great adaptability, it is crucial to pay attention to their specific needs, particularly those related to feeding, parasitic infestations, and stress management [11].

In the EU, the protection of farmed livestock is regulated by Directive 98/58/EC, which does not provide any specific requirements for these species. Similarly, there are no specific provisions on transport duration or minimum space requirements. Some countries have specific provisions for keeping alpacas, such as Switzerland [15] and New Zealand [16], essentially focusing on housing, feeding, and animal handling.

Stress can refer to “conditions where an environmental demand exceeds the natural regulatory capacity of an organism; in particular, situations that include unpredictability and uncontrollability” [17]. Stressors can be classified as psychological or physical [17,18]. Psychological stress can result from social or non-social factors or from controllable or uncontrollable situations. Social stress involves contact with unfamiliar people or separation, while non-social stress includes challenges such as noise, interrupted light-dark cycles, new smells, confined spaces, and immobility. Controllable stress allows the individual to influence the situation (e.g., by hiding), while uncontrollable stress, considered the most impactful in captivity [19], is unavoidable, as in the case of physical restraint.

Physical stressors include disease, hunger, thirst, and extreme weather conditions [17,18,19]. During a stressful event, the hypothalamic–pituitary–adrenal axis activates, releasing substances like catecholamines and glucocorticoids. This triggers a chain of reactions, causing changes in hematology, serum biochemistry, and clinical and behavioral signs. Physiological shifts involve cortisol, blood glucose, creatine kinase, total protein, and blood urea nitrogen [3,10,18,20,21,22]. Clinical findings encompass tachycardia, increased blood pressure, tachypnoea, dilated pupils, hyperthermia, and muscle tension [17]. Behavioural changes manifest as increased alertness, defensive actions, and avoidance, along with enhanced grooming, mounting, suckling, aggression, kicking, vocalization, urination, and defecation [5,22,23].

In animal theme parks, different animal species usually live close together, in relatively small enclosures, in contact with visitors (and their pets), keepers, and other staff. At the same time, these settings offer very favorable living conditions for ectoparasites, thanks to numerous breeding and resting sites and the variety of available hosts [24].

In alpacas, sarcoptic mange emerges as a prevalent and serious condition [11,12,13,25]; moreover, the zoonotic potential of this parasitic disease should not be ignored [26]. Camelids can be infested by mites of the genera *Psoroptes, Sarcoptes, Chorioptes*, and *Demodex*; of these, *Sarcoptes scabiei* var. *cameli* or *auchinae* and *Chorioptes bovis* are the most reported [12,13,25]. Co-infections may also develop [27]. The pattern of intense itching, hair loss, redness, and scabs, especially around the lips and eyelids, strongly indicates sarcoptic mange. Skin lesions on the legs, inner thighs, lower abdomen, armpits, groin, and perineum suggest a potential combination of sarcoptic and chorioptic mange [11]. The constraints regarding treatment choices include: (i) limited response to treatment, requiring a prolonged period for remission of clinical signs; (ii) no licensed acaricides registered for camelids in several countries, including Italy, United Kingdom [12,13], Romania [14], and France [11]; (iii) reduced effectiveness of topical treatments caused by the absence of lanolin in the camelid fiber [28]. Different macrocyclic lactones (avermectins) have been used for treatment [13,14,29]. For these, the oral or topical route of administration has proven to be less effective than the injection route [30], while the combination of ivermectin and amitraz has given some good results [11,12].

However, in none of the published studies was there any reference made to the possible stressful condition the animals may have been in, nor was there any mention of how the animals reacted physiologically and/or behaviourally to the treatment-induced handling.

This case report describes the clinical presentation and treatment of a mixed mite infestation in three alpacas shortly after they arrived at an animal theme park [31,32]. Particular attention was dedicated to stressor identification and management, which was revealed to be crucial for a successful clinical outcome.

## 2. Case Presentation

### 2.1. Case History

In June, four male alpacas aged between five and six months (hereby identified in the text by sequential numbers 1-2-3-4) were purchased from a commercial free-range farm and transferred to an animal theme park located in Northern Italy (about 300 km away from the supplier). The alpacas were housed in a covered 10 × 7 square meters paddock on permanent coconut fiber bedding, with a 3 m long feeding trough and a 3 m long drinking trough. At the first inspection, three days after their arrival, the animals showed aversion towards music, children’s screams, or the barking of visitors’ dogs. They were fearful and refused to be approached and handled during routine examinations. During the following weeks, they continued to show reluctance towards human approach, hid from visitors, and had poor appetite and stunted growth.

In July, alpaca No. 4 suddenly died. Necropsy showed a poor state of nutrition and non-perforated gastric ulcers. After this event, a three-sided covered shelter (3 × 2 m^2^) was placed inside the pen. This proved to be an excellent solution that quietened the alpacas, who used it to hide whenever there were many visitors or whenever visitors got very close to the fence.

In August, the remaining three alpacas always showed an attitude of fear when handled but less discomfort towards visitors. Their appetite had increased, and so had their weight.

In September, the three alpacas were referred for dermatological problems.

### 2.2. Clinical Examination and Diagnosis

Clinical examination of alpaca No. 1 revealed mild itching and the presence of scaly, crusty lesions on the ear pinnae, the sternum, the axillae, the abdominal and perineal regions, and the medial region of the hind limbs. The lesions were accompanied by alopecia, hyperkeratosis, and skin wrinkling (Figure 1).

Clinical inspection of alpaca No. 2 showed a similar impairment, with the presence of crusty lesions on the muzzle, sternum, axillae, limbs, and perineum (Figure 2).

In alpaca No. 3, the lesions were less severe and mainly involved the axillae and the perineal region (Figure 3).

In alpacas No. 2 and 3, no itching was detected. In all animals, the mucous membranes were pink, with a normal refill time.

Macroscopic and parasitological examinations of the feces were performed, and deep skin scrapings were obtained from the costal, axillary, and perineal regions. The coprological exam was negative, while microscopic examination of the skin scrapings revealed the presence of *Sarcoptes scabiei* [33] in all the alpacas (Figure 4).

### 2.3. Treatment and Outcome

All animals were treated with subcutaneous ivermectin (Ivomec^®^, Boehringer Ingelheim Animal Health Italia S.p.A., 35027 Padua, Italy) at the dosage of 0.2 mg/kg on a weekly basis. Skin scrapings were taken at 21 days and about one month after the last treatment (50 days). Pen cleaning and disinfection were performed on day 1 and 21. An assessment of stress was also performed during each clinical examination (Table 1). This included stressor identification, the assessment of behavioral responses (i.e., spitting, ears backward, escape attempts (running/jumping), kicking, collapsing deliberately, vocalization, chasing, urination, trembling), and physiological measures (i.e., tachycardia, tachypnoea, hyperthermia, dilated pupils). The frequency of each behavioral response during clinical examination was scored as follows: (0) never; (1) 1 time; (2) 2–3 times; (3) continuously.

At days 7 and 14, all animals showed no itching, but the lesions had failed to improve.

At day 21, clinical improvement was evident in all alpacas. Rare crusty lesions, but no hyperkeratosis, were detected on the sternum, the axillae, and the medial region of the hind limbs of alpaca No. 1. The same lesions were limited to the axillary region in alpacas No. 2 and 3. The parasitological examination of the skin scrapings taken from the axillary region revealed no presence of *Sarcoptes scabiei* and minimal presence of *Chorioptes bovis* in all animals (Figure 5).

As regards stress assessment, in addition to the stressors that were normally present (e.g., visitors and their pet dogs, children shouting, music, etc.), additional stressors were identified: disease (the mange), novelty (the veterinarian), uncontrollable and unavoidable situations (handling, restraining, medical care, and procedures).

The behavioral responses common to the three alpacas included ears backward, escape attempts (running/jumping), and vocalizations (Table 2). In alpaca No. 3, urination and trembling were also observed on every occasion. Notably, the type and frequency of the behavioral responses did not change or decrease over time.

The physiological responses recorded in the three animals on all occasions were tachycardia (ranging from 120 to 130 bpm) and tachypnoea (ranging from 28 to 35 bpm). Rectal temperature was always normal for all the alpacas (recorded range from 38 °C to 38.7 °C).

Consequently, given the improvement of the clinical conditions but also the persistence of numerous behavioral and physiological signs of severe stress during handling and treatment, it was decided to suspend treatment at day 21, even though the skin scrapings still showed the presence of live parasites. The animals were then inspected once a week, without handling or restraint, to monitor the course of the disease.

One month after the last treatment (day 50), the parasitological exam performed on all the affected skin sites resulted in a negative. Three months later (day 140), no clinical signs were present, the state of nutrition was good, the physiological parameters were normal, and complete hair regrowth was observed in all animals. All three alpacas showed behavioral improvement in the presence of visitors, remaining in the pen area without hiding. However, whenever the veterinarian entered the pen, their behavior changed, showing increased alertness (ears backward), vocalizations, and escape attempts.

## 3. Discussion

Health is a concept that encompasses physical as well as mental and social states. As stated by Broom, disease and welfare are interrelated: on one hand, the welfare of diseased animals is poor, while, on the other hand, “poor welfare, whatever its cause, can lead to increased susceptibility to disease” [34].

Although the influence of stress on human health is generally acknowledged, awareness in the veterinary literature appears to be more limited [35]. Stress is directly connected to a plethora of dysfunctions, such as gastrointestinal function and secretion, intestinal microbiota, immune suppression, pain perception, wound healing, and cutaneous disease [17,35].

The activation of the hypothalamus–pituitary–adrenal (HPA) axis leads to the release of stress-related compounds that trigger diverse immune reactions in the skin [36]. Additionally, skin cells can produce these hormones themselves, as well as their respective receptors, contributing to skin inflammation. For instance, CRH promotes corticosteroid synthesis in dermal fibroblasts and, through ACTH, stimulates the production of corticosteroids in melanocytes and encourages the production of the pro-inflammatory cytokine IL-18, which boosts T-cell activity and supports the generation of Th2 cytokines. Likewise, ACTH stimulates cortisol production, melanogenesis, cytokine production, cell proliferation, dendrite formation, hair growth, and immune and inflammatory regulation. Finally, cortisol contributes to the regulation of hair follicle proliferation and differentiation and the establishment of the epidermal barrier [37]. Besides the HPA axis, the sympathetic nervous system (SNS) also plays a crucial role in managing stress. The activation of the SNS triggers the release of catecholamines, specifically adrenaline and noradrenaline, from the adrenal cortex and sensory nerve endings. Adrenaline interacts with various adrenergic receptors, resulting in reduced blood flow to the skin, decreased fibroblast migration, and collagen secretion, and impaired wound healing. Indeed, since both the skin and the nervous system originate from the embryonic ectoderm, it is clear that there is a strong relationship between these two systems. In fact, if there is a correlation between stress conditions and dermatosis or itching [38], it is also true that mange has a stressful impact on the animal [39].

In our case, the animals were already showing a state of severe stress upon their arrival at the new farm. They were young, they had been separated from their native group and faced a long journey to a new and completely different environment. Moreover, the animal theme park harbored several sources of stress, including music, children’s screams, or the barking of visitors’ dogs, but also the restriction of movement due to the absence of an outdoor paddock [15], the initial absence of a shelter, and the forced proximity to humans [21]. Indeed, chronic stress, resulting in a poor state of nutrition and gastric ulcers, likely caused the death of alpaca No. 4 one month after arrival.

Being a holiday period in Italy, although during August the animal theme park was open, there were few visitors and no school groups; this—together with the shelter set up—was reflected in the alpacas’ increased well-being, as they gained weight and seemed less frightened. However, at the beginning of September, this peaceful situation came to an end, and the animals returned to their previous state of stress.

Clinical management of mange requires a full examination, repeated skin scrapings, and often “invasive” treatment. Indeed, capture, restraint, and forced lying position for at least 15–20 min, even in animals used to being handled, is enough to induce stress in alpacas [20]. Moreover, skin scrapings and injections, although performed under good veterinary practice, contribute to the animals’ discomfort. An attempt was made to stress the animals less by performing the procedures in the same way, at the same time, and by capturing the animals in the same order. Nonetheless, the three alpacas showed altered physiological and behavioral responses throughout the observation period. Alpaca No. 3 seemed to be the most stressed, with symptoms such as trembling and urination. Indeed, this attitude and the release of fear pheromones with urine [18] could have resulted in emotional contagion, influencing the welfare of the group [40]. For these reasons, measuring cortisol or other parameters such as creatine kinase, total protein, blood urea nitrogen, and glucose as indicators of stress [3,10,18,20,21,22] was ruled out. These methods would have required invasive procedures (e.g., blood collection on top of the physical restraint), which are stressful in themselves and not fully justifiable from a clinical point of view. They could have favored the misinterpretation of results [39] and eventually confounded the clinical assessment of the patients [4].

Although numerous substances have been used for mixed infestations, a standardized treatment protocol has not yet been established in alpacas. Successful outcomes have been achieved by the combined administration of subcutaneous and topical ivermectin or subcutaneous ivermectin and topical fipronil [41]. Even the use of a combination of two acaricides (amitraz and ivermectin) with a chlorhexidine shampoo proved to be satisfactory [11]. However, treatment with this type of protocol requires considerable organization and manpower, as well as prolonged handling and an additional novelty (bathing) for the animals, which would have further aggravated their stressful condition. The protocol chosen in this case had already proved effective in llamas, where the mixed infestation of sarcoptic and chorioptic mites was effectively managed with the administration of ivermectin at a dosage of 0.2 mg/kg on a weekly basis for four treatments [42].

On these grounds, considering that the animals persisted in their stressful status, although the symptoms had not completely remitted and the parasitological test results were still positive, it was decided to give them some time, peace, and quiet (as far as possible) to overcome the illness, while continuing to monitor them at a distance.

Regarding the possible origin of the infestation, the alpacas might have already been infested at the farm of origin, and the stress of transport, change of environment, etc., might have favored the clinical onset. Alternatively, they might have become infested in their new environment. Considering that the lifecycle of these mites is about three weeks, that they do not survive more than 3–4 weeks in the environment [39], and that the disease became clinically manifest later than two months after their arrival, the second hypothesis seems to be more likely. However, in both cases, the low level of well-being (i.e., stress) to which the animals were exposed certainly played an important pathogenetic role. This is why the identification and management of stressors should be a fundamental and integral part of therapy, especially in animal species particularly susceptible to stress, such as alpacas.

## 4. Conclusions

This case highlights the key role of stress in the onset and evolution of dermatological diseases and the clear relationship between welfare and disease. The proposed management provides useful information for veterinary practitioners who may find themselves in the same situation, a possibility that is likely to increase over time, given the rapid expansion of these animals. Further studies exploring different, not only medical, but holistic treatment options are urgently needed to validate this approach.

## Figures and Tables

**Figure 1 vetsci-11-00587-f001:**
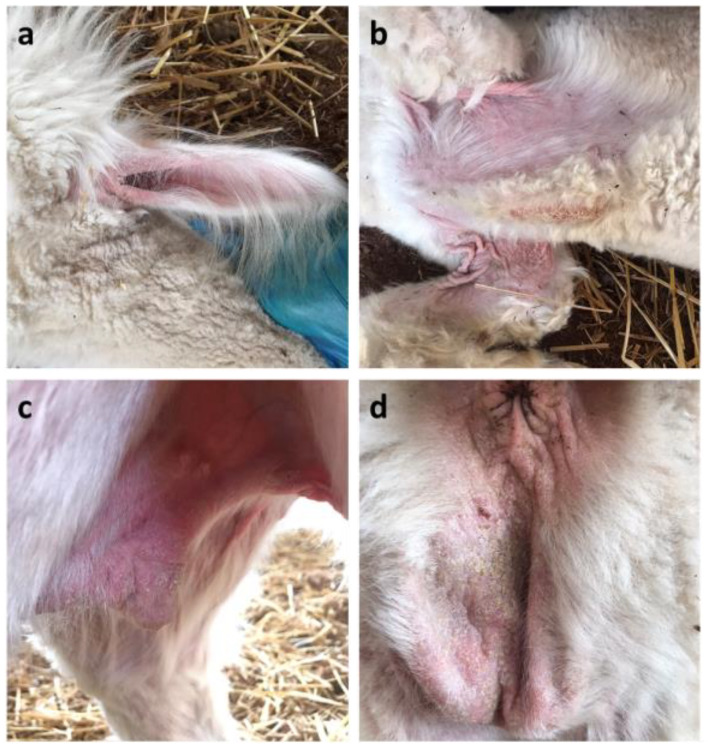
Clinical presentation of alpaca No. 1. Particular of the crusty lesions located on the (**a**) ear pinnae, (**b**) sternum and axillae, (**c**) abdominal region, and (**d**) perineal region.

**Figure 2 vetsci-11-00587-f002:**
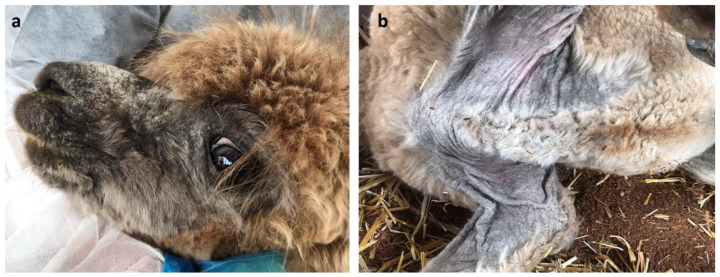
Clinical presentation of alpaca No. 2. Particular of the crusty lesions located on the (**a**) muzzle and (**b**) sternum and axillae.

**Figure 3 vetsci-11-00587-f003:**
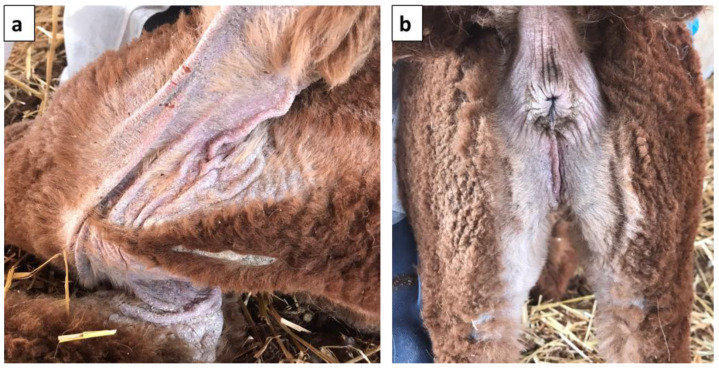
Clinical presentation of alpaca No. 3. Particular of the crusty lesions located on the (**a**) sternum and axillae and (**b**) perineal region.

**Figure 4 vetsci-11-00587-f004:**
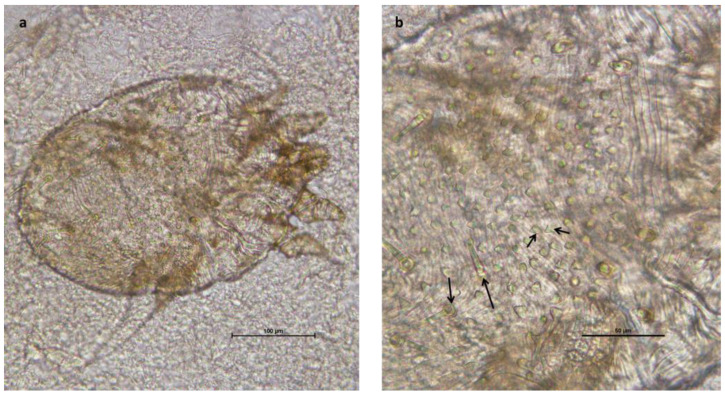
(**a**) Total view of a *Sarcoptes scabiei* specimen (scale bar 100 µ) and detail of the dorsal surface showing dorsal spines (long arrows); (**b**) scales (short arrows) and coarse cuticular striations (scale bar 50 µ).

**Figure 5 vetsci-11-00587-f005:**
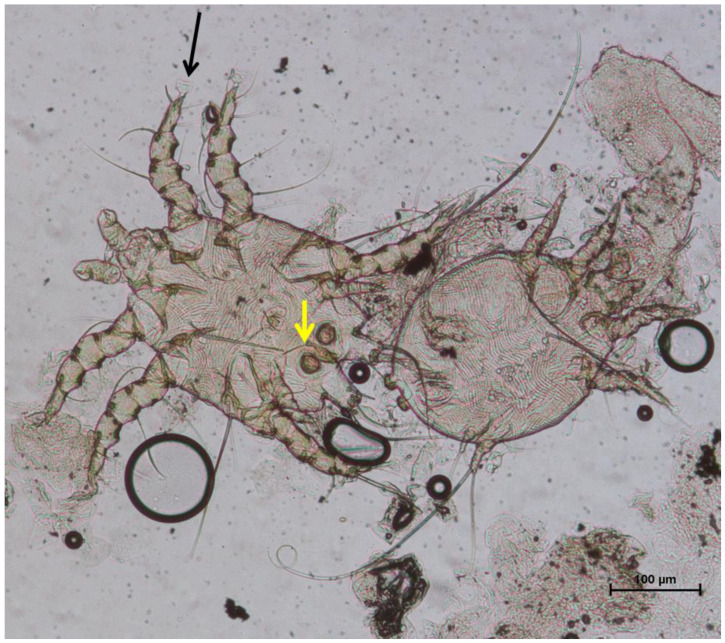
Mating male (left) and female (right) specimens of *Chorioptes bovis*. Black arrow: pulvillus; yellow arrow: copulatory suckers (scale bar 100 µ).

**Table 1 vetsci-11-00587-t001:** Timeline of the interventions.

	Day 0	Day 1	Day 7	Day 14	Day 21	Day 50	Day 140
Clinical examination	✓	✓	✓	✓	✓	✓	✓
Stress assessment		✓	✓	✓	✓	✓	✓
Parasitological examination	✓				✓	✓	
Treatment		✓	✓	✓	✓		
Pen cleaning		✓			✓		

**Table 2 vetsci-11-00587-t002:** Behavioral responses were recorded in the three alpacas. The type and frequency of the responses did not change or decrease over time.

Behavioural Response	Alpaca No. 1	Alpaca No. 2	Alpaca No. 3
Spitting	0	0	0
Ears backwards	3	3	3
Escape attempts	3	3	3
Kicking	0	0	0
Collapsing deliberately	0	0	0
Vocalisation	2	3	3
Chasing	0	0	0
Urination	0	0	2
Trembling	0	0	3

(0) never; (1) 1 time; (2) 2–3 times; (3) continuously.

## Data Availability

The datasets and the isolates analyzed during the current study are available from the corresponding author upon reasonable request.

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
