# Peer review of "Stress Reduction in Alpaca (Vicugna pacos) Mange Management"

_vetsci, 2024, doi:10.3390/vetsci11110587_

Round 1
Reviewer 1 Report
Comments and Suggestions for Authors
This is an interesting paper on alpacas. I have some minor suggestions.
In the methods include what the stress identification behaviours and physiology were that were recorded.
What did you chose ivermectin, why 0.2mg/kg, and why weekly? Why not moxidectin?
The handling seemed way more than was required. Why were they handled is this way and for this period of time.
Author Response
Dear Reviewer,
Thank you for revising our manuscript. We really appreciate your kind comments. We revised our manuscript in line with your suggestions and we hope that this edited version will meet your requirements. Point-by-point responses to your comments are in the attached PDF file.

Reviewer 2 Report
Comments and Suggestions for Authors
See attached file

Author Response

(The authors gave the same response as above.)

Reviewer 3 Report
Comments and Suggestions for Authors
The manuscript titled “Stress reduction in alpaca (Vicugna pacos) mange management” is interesting and well written, but there are some important issues to be addressed in my opinion. First, if the main objective of this study is to assess stress in the management of mangy alpacas, you should add more details on how you assessed stress factors and behaviour in these animals. I suggest to add a stress score (eg from 0 to 5) for each alpaca and put it in a Table describing all the clinical/parasitological factors schematically.
Then the authors should be more detailed when describing the timing of treatment and parasitological exams. You may add a timeline as a figure, eg.
Please find below some line-by-line comments.
Line 20: add a comma between stress and treatment
Line 25: delete comma between mange and caused
Line 28: technically, alpacas are domesticated livestock species, while vicuñas and guanacos are their wild counterparts. Please use another term (e.g., “fragile”) or delete “non-domesticated”.
Line 47: since here you talk about capture myopathy in non-domesticated prey species, I suggest adding some more references on wild ungulates, rather than only ref. 8 which is referred to a single case on an alpaca.
Line 95: the sentence is not clear to me. You may add “THERE was any reference made…”
Lines 107-117: I do not follow how these paragraphs are related to the mange cases described later. If not justified, I suggest deleting or shortening them.
Line 140: The coprological exam
Line 147-149: treatment was administered to which animals? Please specify
Line 168-169: Chorioptes was observed in all animals? In the same regions? Please specify
Lines 173-177: when was the treatment interrupted, after day 21 (after third injection)? Please specify the temporal flow.
Line 195-6: Be more detailed: “diverse immune reaction”, which reactions? Which compounds?
At what time was the last skin scraping done? Were the mites still alive or dead? Please spcify
This would be an important information to add and discuss, as the life cycle of these mites is generally around 10-14 days, so, for example, if the skin check after the last treatment was done after 1 week, it could be that you still found some dead mites, but it doesn’t mean that the treatment didn’t work.
General comment for case description and results: it would be very helpful for a reader to add a Table to describe schematically each alpaca’s clinical condition, treatment and parasitological check.
Moreover, in line 150-152 and 157-158 you explain very shortly how stress was assessed on the alpacas. Nonetheless, since the main objective of your paper is to assess stress in the management of mangy alpacas, you should add more details on how you assessed stress factors and behaviour in these animals. I suggest to add, if possible, a stress score (eg from 0 to 5) that you scored for each alpaca and put it in the Table.
Author Response

(The authors gave the same response as above.)

Round 2
Reviewer 2 Report
Comments and Suggestions for Authors I think the manuscript, so modified, is acceptable for publication. The authors responded sufficiently to requests for clarifications and changes